# Familiarity of an environment prevents song suppression in isolated zebra finches

Anja T. Zai[1,2], Diana I. Rodrigues[1,2], Anna E. Stepien[1,2], Corinna Lorenz[1], Nicolas Giret[3], Iris Adam[4], Richard H. R. Hahnloser[1,2]*

1 Institute of Neuroinformatics, University of Zurich and ETH Zurich, Zurich, Switzerland, 2 Neuroscience Center Zurich (ZNZ), University of Zurich and ETH Zurich, Zurich, Switzerland, 3 Institut des Neurosciences Paris Saclay, UMR 9197 CNRS, Université Paris Saclay, France, 4 Department of Biology, University of Southern Denmark, Denmark

* rich@ini.phys.ethz.ch

**Data Availability Statement:** All relevant data are within the paper and its Supporting Information files.

**Funding:** This study was funded by Swiss National Science Foundation and the European Research

## Abstract

Despite the wide use of zebra finches as an animal model to study vocal learning and production, little is known about impacts on their welfare caused by routine experimental manipulations such as changing their social context. Here we conduct a post-hoc analysis of singing rate, an indicator of positive welfare, to gain insights into stress caused by social isolation, a common experimental manipulation. We find that isolation in an unfamiliar environment reduces singing rate for several days, indicating the presence of an acute stressor. However, we find no such decrease when social isolation is caused by either removal of a social companion or by transfer to a familiar environment. Furthermore, during repeated brief periods of isolation, singing rate remains high when isolation is induced by removal of social companions, but it fails to recover from a suppressed state when isolation is induced by recurrent transfer to an unknown environment. These findings suggest that stress from social isolation is negligible compared to stress caused by environmental changes and that frequent short visits of an unfamiliar environment are detrimental rather than beneficial. Together, these insights can serve to refine experimental studies and design paradigms maximizing the birds' wellbeing and vocal output.

## Introduction

Song production and learning in songbirds show many similarities with human speech learning [1, 2]. A commonly used songbird species in vocal learning research is the zebra finch (*Taeniopygia guttata*) because it is small, easy to keep and breed in captivity, and it has a simple but stereotyped song that birds also produce while alone. Many zebra finch experiments require controlled conditions of prolonged social isolation, for example to achieve high song recording quality or to study undirected (socially isolated or not female-directed) song [3–6]. Because zebra finches are gregarious by nature, it is generally believed that to socially isolate them is burdensome, yet little is actually known about the welfare impact of social isolation.

The literature on corticosterone, a steroid hormone involved in regulation of energy, immune reactions, and stress, provides no clear view on the burden associated with social

Council (ERC). The funders had no role in study design, data collection and analysis, decision to publish, or preparation of the manuscript.

**Competing interests:** The authors have declared that no competing interests exist.

isolation. [7] showed that one day after social isolation, male zebra finches exhibit a 64% increase in corticosterone compared to a control group held in pairs in the same experiment chamber. However, a different study [8] found a return to baseline corticosterone levels just 30 minutes after isolation and to even lower levels than in the control group of [7], suggesting the presence of an uncontrolled and thus unreported manipulation that either amplified or alleviated stress in one of these studies.

Independently, [9, 10] found no difference in corticosterone levels between birds that were isolated either alone or in pairs. In both males and females, corticosterone levels increased during both solo and duo housing compared to control birds that remained in the aviary. That stress in manipulated birds was similarly elevated regardless of the social context in which they were removed from the aviary suggests that perhaps stress is caused by the transfer away from the aviary into a new environment rather than by the social isolation itself. Indeed, handling zebra finches and moving them to a new space already causes an acute increase in corticosterone in (Banerjee and Adkins-Regan, 2011).

Stress in males can also be measured with a behavioral read-out, namely via their singing rate. Many studies have suggested that singing is a positive sign of wellbeing [11–15]. Indeed, singing rate can be a very reliable indicator of recent stress because without presence of a stressful manipulation such as a surgery or tethering, male zebra finches sing at consistent high rates: in 26 isolated birds examined on more than 1000 days, no bird ever sang less than 500 song motifs per day unless stressed [15]. Typically, when a bird is transferred from the colony to an experimental recording chamber where it is kept without social contact, his singing rate is initially very low, below 500 motifs per day, before it gradually increases [15]. These observations suggest that either social isolation, a change in environment, or both constitute a stressor.

To disentangle stress associated with a change in environment from stress caused by social isolation, we perform a post-hoc study of data aggregated in our labs on continuously recorded male zebra finches. We analyzed song data from the preparatory periods of experiments investigating the neural basis of song control or birds' behavioral and learning mechanisms. Birds were transferred both between social and isolate housing conditions and across familiar and unfamiliar environments. Because there are no corticosterone measurements available from these aggregated data, we assess stress in males via their singing rates, seeking clues about welfare implications of social isolation versus changes in the environment (none of the experiments was aimed at manipulating or measuring singing rate).

## Results

To assess how isolation in an unfamiliar environment impacts the singing rate in adult male zebra finches, we analyzed data from males that were moved for several days from same-sex housing in the aviary with auditory contact to females to complete isolation in a sound-proof chamber (see Methods, Fig 1A). For n = 8 naïve-solo birds, the chamber was unfamiliar whereas for n = 8 familiar-solo birds the chamber was familiar because they had previously been isolated (on average 121 ± 142 days before, range 4–444 days) in the same or a similar chamber. If stress arises from isolation per se, there should be no difference in singing rate between these bird groups.

On the first day of isolation (day 0), the naïve-solo birds produced a median of 50 song motifs (range 0–419 motifs), almost 20 times fewer than the median 930 song motifs the familiar-solo males produced on the first day they were back in isolation (range 0–6192 motifs, p = 0.04, Wilcoxon rank sum test, stat = 48.5, Fig 1C).

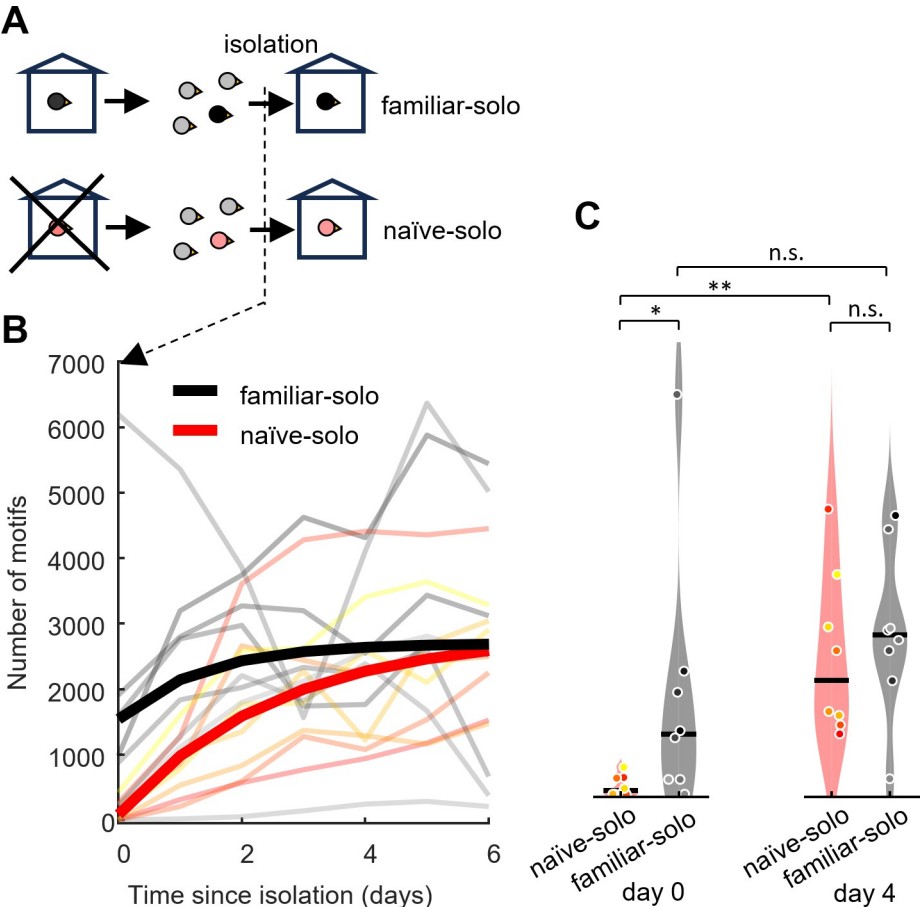

**Fig 1. From aviary to chamber.** High singing rate on first day in familiar environment. **(A)** Familiar-solo birds have been isolated before whereas naïve-solo birds have never seen the isolation environment before. Birds are housed with other males in the colony before being isolated (day 0). **(B)** Daily number of song motifs produced since beginning of isolation. Thin lines represent motif numbers in individual birds and the thick lines represent the trend represented as an exponential fit (n = 8 naïve-solo, yellow to red; n = 8 familiar-solo, gray to black). **(C)** Violin plot comparing the numbers of song motifs produced on day 0 (same groups and colors, * indicates p < 0.05, ** indicates p < 0.01, and *** indicates p < 0.001, Wilcoxon signed rank test, line shows median). Familiar-solo birds produced initially (on day 0) more song motifs than naïve-solo birds.

All n = 8 naïve-solo birds were likely stressed because they produced fewer song motifs than the welfare threshold of $W_{99.5} = 765$ motifs as defined in [15]—a singing rate below this threshold corresponds to 0.5% probability that a bird is diagnosed as stressed even though there is no acute stressor present. By contrast, using the same criterion, only n = 3/8 familiar-solo birds were stressed on day one. These findings show that familiarity of an environment can alleviate stress arising from social isolation.

The singing rate in both naïve-solo and familiar-solo birds gradually increased on subsequent days. By day 4, naïve-solo birds, produced a median of 1758 daily motifs (range 940–4414 motifs), which exceeded their singing rate on day one by a factor of more than 30 (p = 0.008, Wilcoxon signed rank test, stat = 36, n = 8 birds, Fig 1B and 1C), suggesting their successful coping with the new situation. In familiar-solo birds, the motif count on day 4 was even larger, 2468 motifs (range 241–4314 motifs), which however was not different from the count on day 0 (p = 0.08, Wilcoxon signed rank test, stat = 31, n = 8 birds, Fig 1B and 1C). On that day (day 4), there was no significant difference in singing rate between naïve-solo and

familiar-solo birds any more (p = 0.65, Wilcoxon rank sum test, stat = 63, n = 8 birds in each group, Fig 1B and 1C), suggesting that it takes less than a week for birds to adapt to a new environment.

Our analysis reveals a strong effect of familiarity with an environment on the initial singing rate in that environment. But when does an environment become familiar? Is familiarity determined by the total time previously spent there? We found no support for this idea, because in the 8 familiar-solo birds, we found no correlation between the duration of the first-time isolation (median: 12 days, range 2–64 days) and the number of song motifs produced on the first day of re-isolation: r = -0.20, p = 0.63, n = 8 birds. As a caveat, note the diverse time periods between the two isolations (median: 90 days, range 4–444); however, we did not find a correlation between the period since first isolation and the singing rate on the first day of re-isolation (r = -0.03, p = 0.94, n = 8 birds), with the bird singing more than 6000 motifs upon re-isolation having spent 176 days between isolations in the aviary, which was longer than the group median. Thus, familiarity can be reached within several days and it can last for half a year.

These results suggest that the initially low singing rate in naïve-solo birds was mainly due to lack of familiarity of the environment rather than to social isolation. We found further support for this hypothesis in birds that were isolated for the first time in their life without experiencing a change in environment. We analyzed data from adult males that were not housed in the aviary initially, but in mixed-sex pairs inside a recording chamber (they had been removed together from the aviary or been raised in the recording chamber). The males were then isolated in their chamber by removal of the female. If singing rate is suppressed specifically when males are isolated in a new environment, then no suppression should be seen in these isolates. Indeed, after removal of the females, the n = 13 same-solo birds even increased their median singing rate from 1720 motifs (range 0–3675 motifs) on day -1 to 2830 motifs (range 0–6264) on the isolation day 0 (p = 0.002, signed rank test, stat = 2, n = 13 birds, Fig 2A), unlike naïve-solo birds. Thus, first-time isolation can be associated with increases in singing rate, depending on where a bird is isolated.

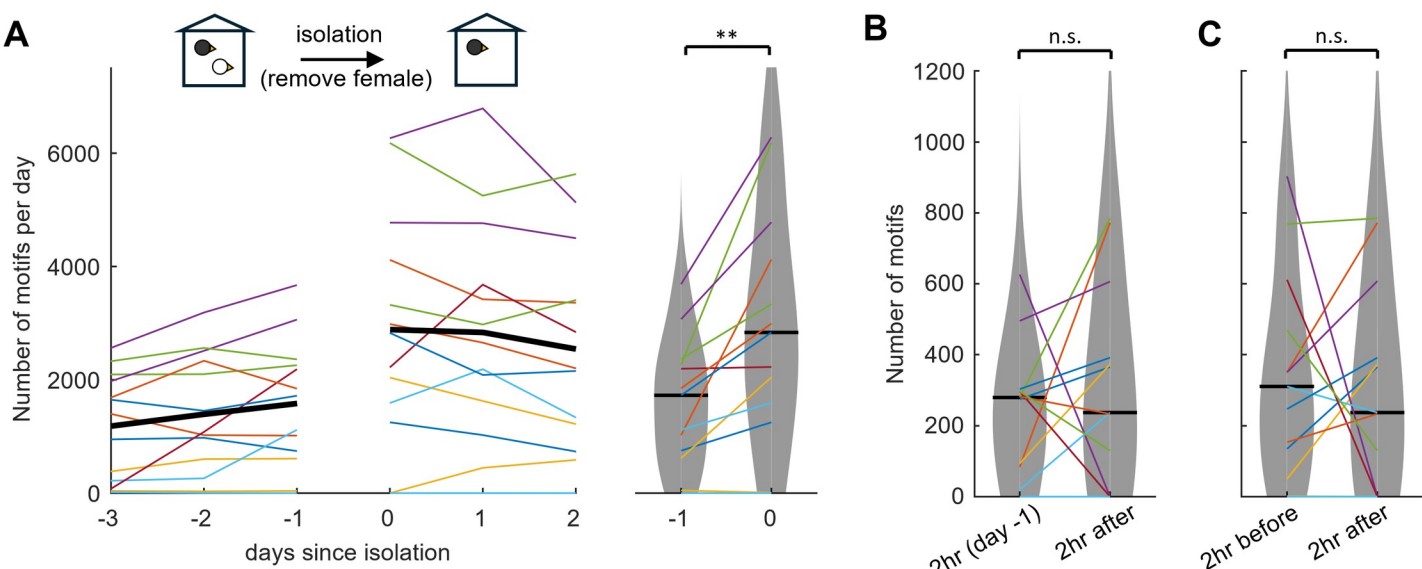

**Fig 2. Social change: No decrease in singing rate upon first-time social isolation in the same environment.** *(A)* Same-solo birds increase their singing rate after first-time isolation caused by removal of the female companion on day zero. Violin plots summarize the singing rates on the days before (day -1) and after isolation (day 1). ** indicates p < 0.01, Wilcoxon signed rank test. No decrease in singing rate is observed in same-solo birds in the first 2-h after social isolation, neither when compared to the same 2-h period on the day before (2-h before) *(B)* nor to the 2-h period before isolation *(C)*. Colors indicate individual birds.

Not even in the first two hours after isolation was there a decrement in singing rate relative to either the same two-hour period on the day before (p = 0.46, signed rank test; 2-h median day -1: 279 motifs, range 0–627; 2-h median day 0: 237 motifs, range 0–785; stat = 24, n = 13 birds, Fig 2B) or the last two hours before isolation (p = 0.85, signed rank test; median 2-h before on day 0: 310, range 0–904, stat = 36, n = 13, Fig 2C), which is remarkable because removal of the female requires chasing and manually capturing her, which can be stressful for both birds.

Same-solo birds sang 70 times more than did naïve-solo birds on the first day of isolation (difference in medians: 2780 motifs, p = 0.004, stat = 183.5, n = 13 same-solo and n = 8 naïve-solo). Based on the 99.5% welfare threshold, only 2 out of 13 same-solo birds were stressed on the first day of isolation. These birds sang on average during the last 4 days before isolation only 7 daily motifs (range 0–16) and 58 daily motifs (range 29–120), suggesting they were stressed already before isolation.

In combination, removal of a companion induces anything but a decrease in singing rate, suggesting that singing is a robust behavior in zebra finches that are isolated without a concurrent change in environment. Overall, these striking results suggest a very small impact on welfare of social isolation.

Can familiarity be gained from repeated short visits of an environment? To characterize the familiarity of an environment as a function of the number of times a bird was transferred there, we studied a group of birds that were isolated for a few hours each day. The n = 14 transient-solo-naïve birds were socially housed together with another male and were transiently isolated in an initially unfamiliar environment during 5 consecutive days for up to 4 hours per day. We argued that if time since first visiting the environment dictates familiarity, then the daily singing rate in transient-solo-naïve birds should increase similarly to that in naïve-solo birds in Fig 1 that were chronically isolated for a week. But we found no support for this notion of familiarity.

On the first day of isolation, transient-solo-naïve birds displayed very low singing rates (median 0 motifs/h, range 0–1, Fig 3A), comparable to that of naïve-solo birds on the first day of isolation (0% bootstrap samples with p < 0.05, Wilcoxon rank sum test; average median over bootstrap samples: 0 motifs/h, range 0–0, see Methods). But on the following days, transient-solo-naïve birds barely increased their singing rate. On day 5, they still produced a median of 0 motifs/h (range 0–22), which was dramatically lower than the 181 motifs/h (average median over bootstrap samples, range 44–373) produced by naïve-solo birds on the same day (100% of bootstrapped sets of 1000 samples each yielded p < 0.001, max p = 4.8 x $10^{-4}$, Wilcoxon signed rank test, Fig 3B), revealing that familiarity is not simply determined by the time elapsed since first visiting an environment and that frequent and short visits are not conducive to gaining familiarity.

Further study confirmed that song suppression in transient-solo-naïve birds was due to the environment and not due to the short periods of social isolation. Namely, we analyzed data from another group of transient-solo males that were continuously housed in a familiar sound-proof chamber (they lived in the chamber for at least 14 days before the start of the experiment) and that were isolated by removal of their two female partners for up to 4 h per day on 5 usually non-consecutive days (Fig 3C). These transient-solo-same birds (n = 7), despite their frequent isolation, produced on the 5[th] day of isolation (day 4) a singing rate of 162 motifs/h (range 66–635), which was significantly higher than that of transient-solo-naïve birds on the same day (p = 0.0002, Wilcoxon signed rank test, stat = 126, n = 7 transient-solo-same birds and n = 14 transient-solo-naïve birds, Fig 3B). The higher singing rate of transient-solo-same birds was already apparent after the first 2 hours of isolation, when they produced 252 motifs/h (average over isolation events, median over birds), compared to 1 motifs/h in transient-solo-

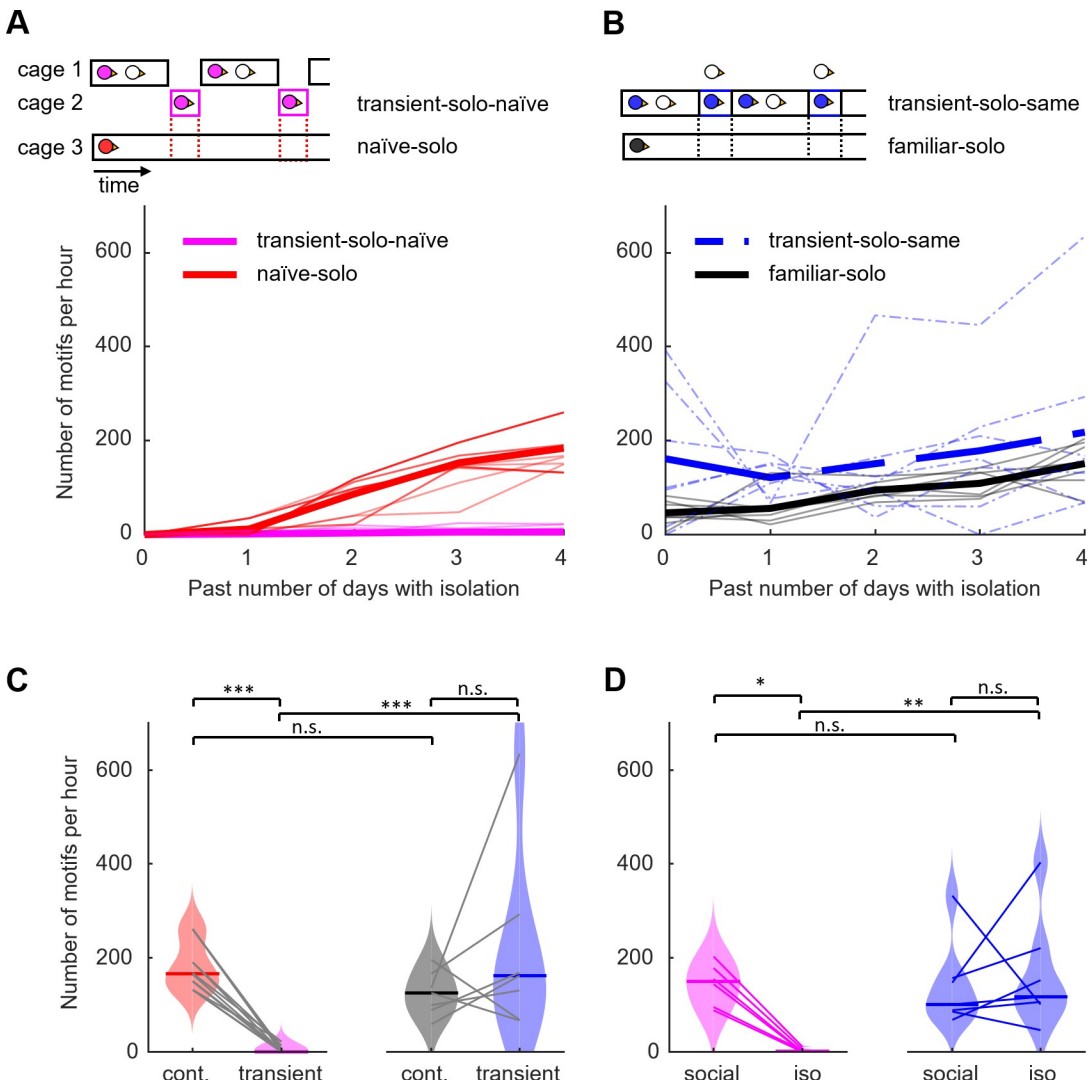

**Fig 3.** Familiarity of an environment is not gained from frequent short visits: **(A)** Number of song motifs produced per hour during isolation in a new environment. Birds shown in magenta (n = 14, 'transient-solo-naïve') were transiently isolated in a previously unknown environment. Red lines show for each pair of transient-solo-naïve birds with the same isolation times the expected number of motifs per hour bootstrapped from naïve-solo birds (n = 7, bootstrapped from Fig 1, 'naïve-solo') that were continuously isolated after moving them from social (colony) to isolation in a new environment. To account for fluctuations in singing rate during the day, continuously isolated birds were time matched to transiently isolated birds and motif rate was measured during the same time period as for the transiently isolated birds before bootstrapping. Thin lines show individual birds and thick lines their average. **(B)** Number of song motifs produced per hour during isolation in the same environment. Birds shown in blue (n = 7, 'transient-solo-same') were transiently isolated for up to three hours per day on five non-consecutive days (dashed) by removing their female companions from their sound isolation chamber. The black lines show the expected number of motifs per hour bootstrapped from solo-familiar birds (n = 7, bootstrapped birds from Fig 1 'solo-familiar') that were moved from social (colony) condition to continuously isolated condition in an environment they have been previously exposed to. As in A, song motifs produced during the same time period were used from the continuously isolated birds for comparison. **(C)** Comparison of the number of song motifs per hour on the fifth day (day 4) of isolation between all four groups of birds shown in A and C (same colours as in A and C). The centre line of the violin plot represents the median over all birds. Grey lines connect birds-bootstrapped samples that were time matched together. *** indicates p < 0.001. **(D)** Comparison of average song numbers before (last 2 h, social) and during isolation (first 2 h, iso) for transiently isolated birds (n = 7 transient-solo-same; n = 6 transient-solo-naïve). Coloured lines connect data points from the same bird. * indicates p < 0.05 and ** indicates p < 0.01, Wilcoxon rank sum test.

naïve birds (p = 0.001, Wilcoxon rank sum test; stat = 70, n = 7 transient-solo-same birds and n = 6 transient-solo-naïve birds, only a subset of birds analyzed, see Methods, Fig 3D), providing further support that a high singing rate depends on the familiarity of an environment rather than the social condition.

Social isolation did not reduce the singing rate of transient-solo-same birds. These birds produced similar numbers of song motifs in the first 2 h after isolation as in the last 2 h just before isolation (p = 0.47, Wilcoxon signed rank test, before isolation 101 motifs, range 68–333; after isolation 117 motifs, range 46–404, stat = 9, n = 7, average over the first 5 isolation events per bird, Fig 3D), which confirms the increased singing rate following separation shown in Fig 2. By contrast, in transient-solo-naïve birds, we observed a significant reduction in singing rate from 149 motifs (range 87–202) during the last 2 h before isolation to 1 motif (range 0–10) during the first 2 h of transient isolation (p = 0.03, Wilcoxon signed rank test; stat = 21, n = 4 isolation events in n = 6 birds, only a subset of birds analyzed, see Methods, Fig 3D).

We found no support for the idea that the difference in singing rates between transient-solo-naïve and transient-solo-same birds arose from a bias in animal selection. Although we selected these two animal groups consecutively without random scheduling, their median singing rates were identical during the last 2 hours before isolation (p = 0.44, Wilcoxon rank sum test; median transient-solo-naïve 149 motifs/h, range 88–202; median transient-solo-same 101 motifs/h, range 73–271, excluding first-time isolation, stat = 43, n = 13 birds, Fig 3D), revealing an equal singing tendency before the treatment.

Given that first exposure to an unfamiliar environment almost completely suppresses song production when the new environment is experienced alone, we wondered whether singing rate is also suppressed when an unfamiliar environment is experienced as a pair. We inspected birds that were socially housed both before and after the transfer to the new environment (Fig 4A). The n = 15 male naïve-duo birds were taken from the aviary and placed with the company of a female into a sound-proof chamber that was unfamiliar to both birds. On the first day (day 0) in the new environment, the naïve-duo males produced a median of 440 motifs (range 0–1680), which was more than 8 times the motif count of naïve-solo birds on the first day (p = 0.03, Wilcoxon signed rank test, stat = 215, n = 8 naïve-solo, n = 15 naïve-duo) but significantly smaller than the motif count in same-duo birds housed with a female for at least 4 days in a familiar environment (p = 0.02, Wilcoxon signed rank test, stat = 135, n = 15 naïve-duo and n = 6 same-duo, Fig 4C, see Methods). By day 4, the singing rate of naïve-duo birds increased to 1108 motifs/day (range 473–2349 motifs/day, n = 11 birds still recorded on day 4, p = 0.001, Wilcoxon signed rank test, stat = 66, n = 11birds, Fig 4B and 4C), comparable to the singing rate of same-duo birds (p = 0.88, Wilcoxon signed rank test, stat = 101, n = 11 naïve-duo and n = 6 same-duo, Fig 4C). Thus, a female companion is conducive to early singing in a new environment, but complete adaptation of singing rate takes several days to unfold.

We further validated our findings using a linear mixed model on the combined data from all continuously manipulated groups (excluding transient-solo-naïve and transient-solo-same). Modeling the motif count on the first day after transfer (see Methods) confirmed our findings: There is a significant fixed effect (besides the general offset) on singing rate of the familiarity of an environment (fixed effect 1536 motifs, p = 0.0008, tStat = -3.6, DF = 47). In contrast, the effect of social housing (duo) is not significant (fixed effect -490 motifs, p = 0.18, tStat = -1.3, DF = 47), demonstrating that to isolate a male bird does not decrease its singing rate unless the isolation is enforced in a new environment. Our recommended procedure for isolating birds in a new environment is shown in Fig 5.

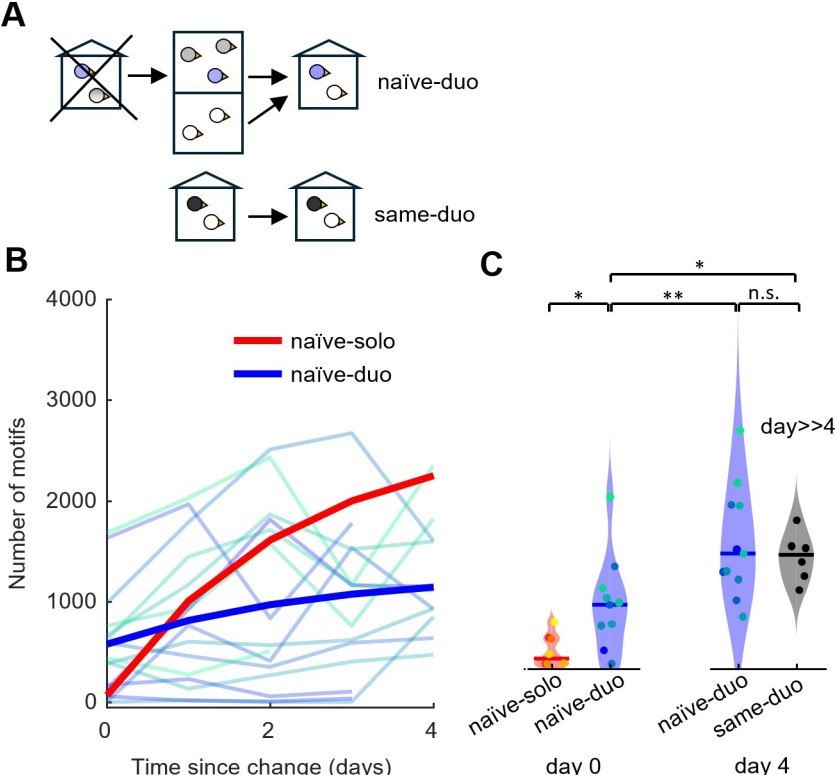

**Fig 4. From aviary to chamber with a female.** A female companion partially mitigates the song suppression in new environments. **(A)** Naïve-duo birds are placed into an unfamiliar sound-proof chamber together with a female. Same-duo birds have been in the familiar environment with their female for more than 4 days. **(B)** Daily number of song motifs produced since transfer to sound-proof chamber. The thin lines represent motif numbers in individual birds and the thick line represents an exponential fit to the group trend ($n$ = 15 naïve-duo, turquoise to blue). The trend of the naïve-solo birds from Fig 1 for comparison. **(C)** Violin plot comparing the numbers of song motifs produced on day 0 and day 4 from naïve-duo (blue), naïve-solo birds on day 0 (red) and same-duo birds (black). Naïve-duo birds produced initially (on day 0) more song motifs than did naïve-solo birds (from Fig 1 for comparison), revealing a song-conducive effect of the female companion (* indicates $p < 0.05$, and ** indicates $p < 0.01$, Wilcoxon signed rank test).

## Discussion

Overall, we found that to transfer a male zebra finch to an unfamiliar environment suppresses singing rate, but social isolation does not, since to isolate a male in the same environment by removal of the female partner has a positive impact on singing rate. Curiously, isolation in an unfamiliar environment is the most song-suppressive manipulation and isolation in the same environment is the most song-stimulating manipulation. In combination, social isolation by itself is unrelated to singing rate.

In experimental designs, recording birds in isolation is often necessary and thus from a welfare perspective, frequent environmental changes and longer periods of isolation have to be weighed against each other. When birds are isolated in a new environment, they increase their singing rate after one day and they attain high singing rates within a few days. But when birds are transiently isolated in a new environment, their singing rate does not recover within 5 days.

Given the positive relationship between singing rate and welfare, transient isolation in unfamiliar environments should be avoided. Namely, if transient isolation can be achieved in the same environment, birds seem to experience no stress, similar to the joy of having the house to

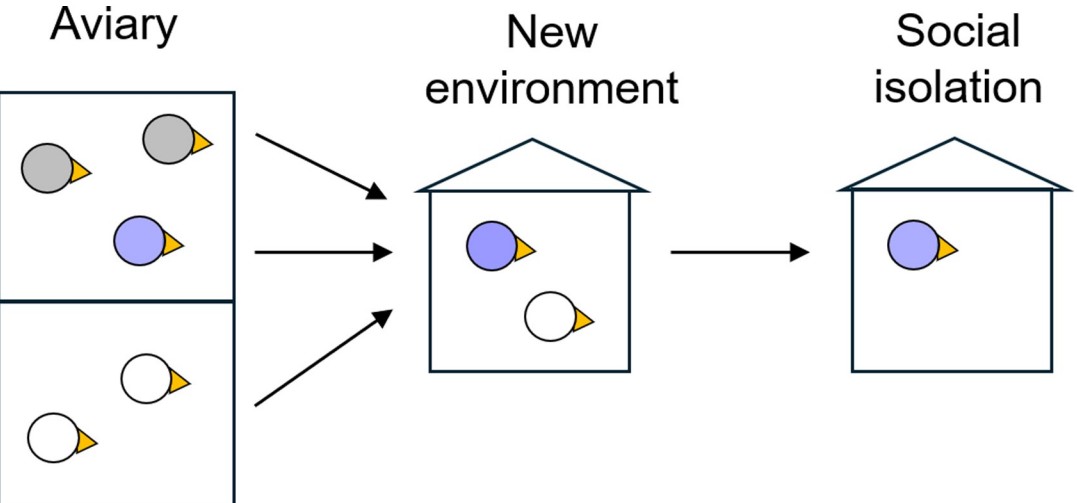

**Fig 5. Our recommendation on how to isolate a songbird in an unfamiliar environment is to first transfer the birds in pairs to the new environment and then to isolate one of them.**

oneself when the family leaves for a trip. Overall, if isolation is required, we recommend transferring birds to a new environment as a pair and then to isolate the male by removing the female partner after a few days.

Our study suggests that the stress-determining factor in previous studies such as [7, 8] was not the social isolation itself but the change in environment. Indeed, in [7] where the stress was higher, the birds were placed in the experimental cage only the day before the experiment and only once so. In contrast, in [8], where the stress was lower, each bird has been tested three times, making it likely that the birds were placed in an unfamiliar environment only once, whereas on the two subsequent tests they were placed in an already familiar environment, possibly explaining the lower corticosterone levels. Our findings agree with [9] in that transfer away from the aviary to a new environment can be stressful, regardless of whether birds are transferred alone or in a pair.

We identified five main limitations of our study that we discuss in the following.

First, our comparison of welfare in single and pair-housed birds in Fig 4 is problematic. Our method of welfare assessment is rooted in statistical (Bayesian) estimation, it is based on subjecting isolated birds to a stressor and to infer the presence of the stressor from the singing rate by comparison to unmanipulated control birds [15]. Our assessment of welfare therefore is appropriate for the isolated birds in Figs 1–4 but not for the socially housed birds in Fig 4. Since a hypothetical link between singing rate and stress has never been measured in pair-housed birds, the generalization to this condition is unclear. It is known that to pair finches with non-preferred mates can dramatically increase corticosterone levels by a factor of 3–4 and over a period of several weeks [16], suggesting that pair housing can be very stressful. To obtain a good baseline measurement of singing rate in pair-housed birds, the mate choice preference therefore needs to be known. Our data is unsuitable to obtain this baseline and our data is potentially biased, since we ignored mate choice preference. We leave it to future work, to measure the singing rate baseline and the effect of stressors in pairs of mutually preferred mates.

Second, our subjects were only males, not females. To extend our work to non-singing females exposed to similar manipulations, one could inspect the production rates of certain calls. For example, distance calls would be interesting to look at, since isolated males produce fewer distance calls when socially isolated [7], suggesting these calls might serve as a read-out

of socially-induced stress. One would need to re-examine a possible confound between social isolation versus unfamiliarity of an environment using these calls, since unlike [7, 17] found that males tend to produce more distance calls when isolated. Similarly, to assess welfare, one could also consider certain acoustic features of calls such as fundamental frequency (pitch) that correlate with stress [7].

Third, we did not randomize subjects (since this is a post-hoc study). However, the many-fold effects we found make it very unlikely that randomization would make a difference.

Fourth, corticosterone measurements would allow testing for possible hormonal implications of our singing-rate measurements and so could serve to further corroborate our findings.

Fifth, juvenile birds would need to be re-examined. Our transient-solo-same birds were juveniles and did not reduce their singing rate when transiently isolated, but the importance of social exposure for normal song learning warrants a more thorough study.

Despite these limitations, our findings provide guidelines for designing vocal-learning studies where high singing rates are key for success. Namely, in addition to the general increased statistical power stemming from larger data sets, many new analyses take full advantage of the high dimensionality of song [18–20], requiring large data sets with many song repetitions, ideally recorded within a short period of time [21]. A temporal separation between transfer to a new environment and imposing social isolation aligns the interests of science and animal welfare.

Finally, our work adds one more clue about why birds sing. According to the conventional theory, male songbirds sing to defend their territories and to attract mates. Zebra finches, however, are not territorial and they form long-lasting monogamous bonds early in life, so theory argues they should not sing much later in life. Recent studies suggested an expansion of the function of song beyond territorial defense and mate attraction to encompass social functions such as pair-bond maintenance and social group cohesion [22, 23]. However, while these social functions explain high singing rates in socially housed adult zebra finches, they fail to explain why birds continue to sing plentiful when socially isolated. A possible explanation has been put forward in that frequent singing during development is important for reaching high muscle speeds [24] and lack of exercise in adults for 2 days already decreases muscle performance and after a week it affects song performance [25]. Isolate song, therefore, could constitute exercise that serves to maintain song. The song suppression in new environments we find then highlights the cost of this exercise: when uncertainty about an unfamiliar environment and its possible dangers prevails, similarly to when stressed or sick, birds take a break from exercising.

## Methods

Animals and housing:

We performed a post-hoc analysis using data recorded prior to ongoing and published [15] studies of n = 77 zebra finches (*Taeniopygia guttata*). The data were acquired for diverse experiment goals that were unrelated to singing rate.

The subjects were all younger than 1155 days at the beginning of the experiment. 65 birds were bred and raised in Switzerland (University of Zurich/ETH Zurich) or France (Université Paris Saclay), where they were kept on a 14/10 hours light/dark daily cycle. 12 birds were raised in Denmark (University of Southern Denmark) where they were kept on a 13/11 hours light/dark daily cycle. All birds were >85 dph old at the time of isolation except for transient-solo-same birds that were between 60 and 90 dph.

Birds had access to food and water ad libitum. Since our manipulations did not involve any painful procedures, we administered neither anesthesia nor analgesia. No animal was killed for the sake of this study. All experimental procedures were approved by the Cantonal Veterinary Office of the Canton of Zurich, Switzerland (license numbers 207/2013 and ZH077/17).

Experiments in Switzerland were carried out in accordance with relevant guidelines and regulations (Swiss Animal Welfare Act and Ordinance: TSchG, TSchV, TVV), experiments in France were approved by the French Ministry of Research and the ethical committee Paris-Sud and Centre (CEEA N˚59, project 2017–12), and experiments in Denmark were approved by the Danish Animal Experiments Inspectorate (Copenhagen, Denmark) and were conducted in accordance with the Danish law concerning animal experiments and protocols.

## Song recordings

We recorded vocalizations with microphones (Audio-Technica PRO 42, Audio-Technica, Tokyo, Japan or Behringer ECM8000) attached to the wall or above the cage. Most of the recordings were song-triggered by detecting rapid sequences of sound segments above a manually chosen harmonicity threshold, making use of a 1-s long data buffer to not miss the first vocalization in the sequence (note that because most recordings were triggered by songs, our data are unsuitable for analyzing the production rate of calls). Other recordings were triggered by supra-threshold sound amplitudes, also making use of a 1-s data buffer. Sounds were digitized to 16 bits at a sampling rate of 32 kHz, except for same-solo birds (44.1 kHz) and transient-solo-naïve birds during transient isolation (16 kHz). Songs were analyzed with custom MATLAB (MathWorks, Inc, Natick MA, United States) scripts. We visualized vocalizations as log-power sound spectrograms (time-frequency representations of sound intensity).

## Bird groups and housing

Before the experiment, birds were housed in the colony in separate-sex social cages (with visual and auditory contact to birds of the other sex). During the experiments, birds were housed either in social cage A (60x50x50 cm$^3$), social cage B (61x61x50 cm$^3$), social cage C (55x32x27 cm$^3$), metal cage (39x23x39 cm$^3$), or plexiglass cage (39x23x39 cm$^3$). Social cages A and B are formed by the interior of sound isolation boxes (the entire isolation box is a cage), as the names imply, in these cages birds are housed socially. All other smaller cages are placed inside similar sound isolation boxes. Birds isolated in plexiglass and metal cages have no contact with other birds. Two metal cages can be joined to form social cage D either with full contact (joining doors open) or no physical contact (joining doors closed).

The bird groups and their housing conditions are detailed in the following.

**A) familiar-solo and naïve-solo (Fig 1).** Male birds (n = 16) were moved from the colony, where they had full contact with other males and visual plus auditory contact with females to a metal cage inside a sound isolation chamber where they were housed alone (without social company). Data from 6/16 birds have been published previously in [15], in that study we did not distinguish between familiar and unfamiliar environments.

Some animals (familiar-solo, n = 8, Fig 1, black) had previously been isolated in a similar environment for another experiment (iso familiar), all other males (naïve-solo, n = 8, Fig 1, red) have never left the aviary and never seen the sound isolation chamber before their isolation. Of the n = 8 familiar-solo birds, during their previous stay in a similar isolation chamber, 3 birds were exposed to white-noise song conditioning triggered by the pitch or duration of a syllable as in [26], the other 5 birds were unmanipulated (only song recordings).

**B) same-solo (Fig 2).** Males (n = 12) were housed in mixed-sex pairs (one male, one female) in cage C for at least 5 days (to familiarize themselves with the cage and the female, two pairs have been raised inside the isolation box). The males were then isolated in their cage for the first time in their life by removing the female no earlier than 2 hours after the light in their isolation chamber went on and no later than 2 hours before the light was switched off.

We measured the singing rate during social housing (with the female) and during isolation (with the female companion removed).

**C) transient-solo-naïve (Fig 3A).** On five consecutive days, males (n = 14) were transiently moved for 1–4 hours per day from social housing in cage B to isolate housing in a plexiglass cage inside a different sound isolation chamber. Before their first isolation, they were housed with a female companion for 5 to 14 days (no data available). After the first and subsequent isolations, they were returned to the same or a similar cage B where they were housed together with a male companion that underwent the same procedure (7 male pairs). We counted songs during isolation in all n = 14 males. In n = 6 males (3 pairs), we additionally counted songs during the social period when they were housed with another adult male (visually discriminating the songs by the two males).

**D) transient-solo-same (Fig 3C).** Young male zebra finches (60–90 dph, n = 7) were housed with two females in a social cage A. On five to eight mostly non-consecutive days (all males < 90 dph), the female companions were moved to a different social cage and the male remained isolated for 2 to 3 hours. Thereafter, the female companions were returned to the male's cage. Vocalizations were recorded and analyzed during the social and isolated housing periods.

**E) Naïve-duo (Fig 4).** Adult males were moved together with a female from the aviary either to a social cage B (n = 13 males) or to a social cage D with full contact (n = 2 males) for 3 to 14 days, vocalizations were continuously recorded.

**F) same-duo (Fig 4C).** Males (n = 6) were housed in a social cage for at least 13 days, spending the last 5+ days together with a female companion. Two males had been raised together with their female sibling in a social cage C and were used in the same-solo group later. Four males have been isolated before: one naïve-solo bird in a social cage D with full contact and three familiar-solo birds in social cage D without physical contact. We counted the males' song motifs on the last day they spent with the female.

Motif counting:

To count the number of song motifs produced by a bird, we selected one song syllable that was present in most motifs and that was easy to detect. Usually, we took the first syllable of the song motif or a harmonic syllable that had been already processed for the originally planned experiment. Whenever possible, we avoided counting syllables that birds repeated in their song motifs. We counted motifs (of familiar-solo, naïve-solo, naïve-duo housed in social cage D with full contact, and same-solo birds) using a neuronal network that detected the target syllable as described in [15], followed by visual verification. In transient-solo-naïve, transient-solo-same, and naïve-duo birds housed in social cage B, we counted motifs using a custom syllable classification and sorting method based on tSNE (t-distributed stochastic neighbor embedding) [27], described in the following.

We first aggregated data from several days. When too much noise (e.g. wing flaps, sounds of birds moving around the cage) was present in the recordings, we removed noise using a K-means algorithm trained on labeled noise examples. We then detected vocalizations using a manually chosen sound-amplitude threshold and compressed the chosen vocalizations down to 400+ principal components. On these, we ran the tSNE algorithm and manually selected the tSNE clusters containing the target syllable, making manual corrections where needed to discard false positive detections.

Although we used diverse methods to count song motifs, this methodological diversity could not account for the observed differences in singing rates. For 3 naïve-solo birds, we counted the motifs twice, using both the neuronal network and the t-SNE method. Over 45 days in total (15 consecutive days per bird), the average absolute difference was 10± 9 motifs

(range 0–36), implying a relative difference of only 0.5%± 0.5% (range 0–1.6%), which is negligible compared to the large effects observed.

## Visualization and statistical tests

To visualize trends for naïve-solo, familiar-solo, and naïve-duo birds, we fitted an exponential function to the data in Figs 1 and 4 using Matlab's fitnlm function. Because no exponential trend was seen in the daily motif counts in transient-solo-naïve (Fig 2) and same-solo birds (Fig 3), we plotted the average count instead in these figures (the median would mask data from one bird).

The violin plots show the singing rates for all birds and the horizontal lines their medians.

To test for statistical significance of differences in medians in paired data (i.e., same birds but different time points), we used the Wilcoxon signed rank test; to test for significance of medians of two independent samples (i.e., different bird groups), we used the Wilcoxon rank sum test. We used the Wilcoxon test because of occasional outlier birds, making the data clearly not gaussian distributed. The value *stat* refers to the Wilcoxon test statistics.

## Comparing transient and continuous isolation

To compare singing rates of transiently versus continuously isolated birds, we chose matched time windows for the latter to avoid biases due to circadian trends (matched isolation periods relative to the lights-on time). We considered only the first 5 isolation events, the common minimum among all birds.

We compared transient-solo-same birds to (continuous) familiar-solo birds by randomly drawing for each transient-solo-same bird one familiar-solo bird (with replacement), resulting in 7 time-matched birds in each group. In matched time windows between these two groups, we did not find a significant difference in motif counts (average p = 0.47, 5% of bootstrap samples with p < 0.05, Wilcoxon signed rank test, Fig 3B).

We compared transient-solo-naïve to (continuous) naïve-solo birds by randomly drawing for each transient-solo-naïve couple a naïve-solo bird (with replacement), resulting in 7 time-matched triplets of birds. We then counted the motifs during the isolation period of transient-solo-naïve birds and during matched time periods in naïve-solo birds and divided the counts by the duration of isolation, repeating this process 1000 times to obtain a representative bootstrap sample. We report the statistics of the continuously isolated group as averages and ranges over all 1000 bootstrap samples (Fig 3). For some random draws, time matching was not possible on the first day of isolation (because continuously isolated birds were isolated later during the day than transiently isolated birds). These cases were ignored in averages for the first day but included in averages for the fifth day.

## Linear mixed model

We compared the number of song motifs on the first day in naïve-solo (Fig 1), familiar-solo (Fig 1), same-solo (Fig 2), naïve-duo (Fig 4), and same-duo (Fig 4) birds. We fitted a linear mixed model on the combined data with fixed effects corresponding to a general offset a, the familiarity of the environment b (b = 1 for familiar-solo, same solo, and same duo), and the presence of a companion c (c = 1 for naïve-duo and same-duo). We further included a random effect for each bird.

## Supporting information

**S1 Data.**
(ZIP)

## Acknowledgments

We thank Maria Westphal Anthonsen and Heiko Hörster for help with recording birds.

## Author Contributions

**Conceptualization:** Anja T. Zai, Anna E. Stepien, Richard H. R. Hahnloser.

**Data curation:** Anja T. Zai, Corinna Lorenz.

**Formal analysis:** Anja T. Zai.

**Funding acquisition:** Richard H. R. Hahnloser.

**Investigation:** Anja T. Zai, Diana I. Rodrigues, Anna E. Stepien, Corinna Lorenz, Nicolas Giret, Iris Adam.

**Methodology:** Anja T. Zai, Anna E. Stepien, Iris Adam, Richard H. R. Hahnloser.

**Project administration:** Richard H. R. Hahnloser.

**Software:** Anja T. Zai.

**Supervision:** Richard H. R. Hahnloser.

**Visualization:** Anja T. Zai.

**Writing – original draft:** Anja T. Zai, Richard H. R. Hahnloser.

**Writing – review & editing:** Anja T. Zai, Anna E. Stepien, Corinna Lorenz, Nicolas Giret, Iris Adam, Richard H. R. Hahnloser.

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
