## [Decision Letter · Decision Letter 0]

23 Aug 2024

PONE-D-24-26690Familiarity of an environment prevents song suppression in isolated zebra finchesPLOS ONE

Dear Dr. Hahnloser,

Thank you for submitting your manuscript to PLOS ONE. After careful consideration, we feel that it has merit but does not fully meet PLOS ONE’s publication criteria as it currently stands. Therefore, we invite you to submit a revised version of the manuscript that addresses the points raised during the review process.  Before we discuss the review, let me apologize for the exceptionally slow processing time on your manuscript.  I have struggled to find reviewers willing to commit their time to this process, and then those that did commit continually delayed responding to the point that we just had to move forward without their contributions to your work.  I sincerely apologize for this delay; I am quite frustrated with our fellow scientists not being more giving of their time. As you will find in the reviewer who provided a thoughtful analysis of your paper is positive about the work, as am I.  However, I similarly wonder about the single measure of welfare being centered on song rate.  I realize that you are using animals from diverse studies and would ask that you consider with the data you have available other metrics or discuss this limitation in your manuscript in more detail.  Please address the comments raised and submit your revised manuscript by Oct 07 2024 11:59PM. If you will need more time than this to complete your revisions, please reply to this message or contact the journal office at plosone@plos.org. Please include the following items when submitting your revised manuscript:A rebuttal letter that responds to each point raised by the academic editor and reviewer(s). You should upload this letter as a separate file labeled 'Response to Reviewers'.A marked-up copy of your manuscript that highlights changes made to the original version. You should upload this as a separate file labeled 'Revised Manuscript with Track Changes'.An unmarked version of your revised paper without tracked changes. You should upload this as a separate file labeled 'Manuscript'.If applicable, we recommend that you deposit your laboratory protocols in protocols.io to enhance the reproducibility of your results. Protocols.io assigns your protocol its own identifier (DOI) so that it can be cited independently in the future. For instructions see: https://journals.plos.org/plosone/s/submission-guidelines#loc-laboratory-protocols. Additionally, PLOS ONE offers an option for publishing peer-reviewed Lab Protocol articles, which describe protocols hosted on protocols.io. Read more information on sharing protocols at https://plos.org/protocols?utm_medium=editorial-email&utm_source=authorletters&utm_campaign=protocols.

We look forward to receiving your revised manuscript.

Kind regards,

Brenton G. Cooper, Ph.D.

Academic Editor

PLOS ONE

Journal Requirements:

3. Thank you for stating the following financial disclosure: Swiss National Science Foundation (Projects 31003A_182638 and 31003A_156976/1) 

European Research Council (ERC) Advanced Grant (268911, VOTECOM)  

4. Thank you for stating the following in the Acknowledgments Section of your manuscript: We thank Maria Westphal Anthonsen and Heiko Hörster for help with recording birds. This

work was supported by the Swiss National Science Foundation (Projects 31003A_182638

and 31003A_156976/1) and the European Research Council (ERC) Advanced Grant

(268911, VOTECOM).

Please remove any funding-related text from the manuscript and let us know how you would like to update your Funding Statement. Currently, your Funding Statement reads as follows: Swiss National Science Foundation (Projects 31003A_182638 and 31003A_156976/1) 

European Research Council (ERC) Advanced Grant (268911, VOTECOM)

Reviewers' comments:

Reviewer's Responses to Questions

**Comments to the Author**

1. Is the manuscript technically sound, and do the data support the conclusions?

Reviewer #1: Yes

2. Has the statistical analysis been performed appropriately and rigorously? 

Reviewer #1: Yes

3. Have the authors made all data underlying the findings in their manuscript fully available?

Reviewer #1: Yes

4. Is the manuscript presented in an intelligible fashion and written in standard English?

Reviewer #1: Yes

5. Review Comments to the Author

Reviewer #1: Zebra finches are commonly used in behavioral and neuro-ethology experiments. In this study, authors investigated how improve the welfare of domesticated zebra finches during experiments that require social isolation. The study aims at separating between isolation stress and novel environment stress. Findings suggest that novel environment is a much stronger stressor. Importantly, novel environment stress can be reduced at least in part, by first housing the birds in pairs for a few days before separating them. Finally, the study found that repeated short exposures to a novel environment are not useful.

Overall, the study is well done and well presented. I only have one concern: The entire study is based on singing rate as an indication of welfare, but I suspect that this is a bit oversimplifying assumption: Clearly, a lack of singing or very low level of singing is an indicator of welfare issue. But we do not know if there is an optimal level of singing rate. Authors assume that more singing is always better. But there is no evidence to that. In fact, they found higher level of singing when they removed the female partner. Maybe this higher level is an indicator of stress or frustration?

My suggestion is to use song levels in the presence of a female partner as a baseline.

6. PLOS authors have the option to publish the peer review history of their article (what does this mean?). If published, this will include your full peer review and any attached files.

Reviewer #1: No

---

## [Author Response · Author response to Decision Letter 0]

11 Oct 2024

We thank the editor and the reviewer for their thoughtful comments.

We agree there is a limitation of our welfare assessment. But it is not true that we assume that more singing is always better. Our measure of welfare is rooted in statistical (Bayesian) estimation. It is based on subjecting isolated birds to a stressor and to inferring the presence of the stressor from birds’ singing rate by comparison to unmanipulated control birds (that were not subjected to a stressor). In this experiment, all stressed birds tended to reduce their singing rate, whence the origin of the welfare threshold (1). In other words, if stressed birds were to sing more, our detection criterion for poor welfare would be a lower bound, not an upper bound. Our assessment of stress in isolated birds in Figures 1-4 is therefore not based on any assumptions about singing rate.

Issues with our welfare assessment only arise when we use it to evaluate the welfare of socially housed birds in Figure 4. Since the link between singing and stress has been measured in isolated birds in (1), the generalization to pair-housed birds is unclear. But to simply use bonded birds as a baseline as the reviewer suggests is problematic. To pair finches with non-preferred mates can dramatically increase corticosterone levels by a factor of 3-4 and over a period of several weeks (2), suggesting that to simply consider our paired birds as a baseline will yield a highly variable and potentially biased baseline since we ignored mate choice preference in the current experiment. We leave it to future work to measure the baseline singing rate and the effects of stressors in pairs of mutually preferred mates. 

We highlight this limitation in the discussion as the first limitation of our work.

References

1. Yamahachi H, Zai AT, Tachibana RO, Stepien AE, Rodrigues DI, Cavé-Lopez S, et al. Undirected singing rate as a non-invasive tool for welfare monitoring in isolated malezebra finches. Plos One. 

2. Griffith SC, Pryke SR, Buttemer WA. Constrained mate choice in social monogamy and the stress of having an unattractive partner. Proc Biol Sci. 2011 Sep 22;278(1719):2798–805.

---

## [Editor Report · Decision Letter 1]

17 Oct 2024

Familiarity of an environment prevents song suppression in isolated zebra finches

PONE-D-24-26690R1

Dear Dr. Hahnloser,

We’re pleased to inform you that your manuscript has been judged scientifically suitable for publication and will be formally accepted for publication once it meets all outstanding technical requirements.

Kind regards,

Brenton G. Cooper, Ph.D.

Academic Editor

PLOS ONE
---

## [Editor Report · Acceptance letter]

7 Nov 2024

PONE-D-24-26690R1 

PLOS ONE

Dear Dr. Hahnloser, 

I'm pleased to inform you that your manuscript has been deemed suitable for publication in PLOS ONE. Congratulations! Your manuscript is now being handed over to our production team.

Kind regards, 

on behalf of

Dr. Brenton G. Cooper 

Academic Editor

PLOS ONE